# Revealing the Phenotypic and Genomic Background for PHA Production from Rapeseed-Biodiesel Crude Glycerol Using *Photobacterium ganghwense* C2.2

**DOI:** 10.3390/ijms232213754

**Published:** 2022-11-09

**Authors:** Irina Lascu, Ana Maria Tănase, Piotr Jablonski, Iulia Chiciudean, Maria Irina Preda, Sorin Avramescu, Knut Irgum, Ileana Stoica

**Affiliations:** 1Department of Genetics, Faculty of Biology, University of Bucharest, 050095 Bucharest, Romania; 2Department of Chemistry, Faculty of Science and Technology, Umeå University, S-90187 Umeå, Sweden; 3Department of Organic Chemistry, Biochemistry and Catalysis, Faculty of Chemistry, University of Bucharest, 030018 Bucharest, Romania

**Keywords:** polyhydroxyalkanoates, genomics, Biolog Phenotypic Microarray, fatty acid metabolism, crude glycerol, bioreactor, biopolymer molecular mass

## Abstract

Polyhydroxyalkanoates (PHA) are promising biodegradable and biocompatible bioplastics, and extensive knowledge of the employed bacterial strain’s metabolic capabilities is necessary in choosing economically feasible production conditions. This study aimed to create an in-depth view of the utilization of *Photobacterium ganghwense* C2.2 for PHA production by linking a wide array of characterization methods: metabolic pathway annotation from the strain’s complete genome, high-throughput phenotypic tests, and biomass analyses through plate-based assays and flask and bioreactor cultivations. We confirmed, in PHA production conditions, urea catabolization, fatty acid degradation and synthesis, and high pH variation and osmotic stress tolerance. With urea as a nitrogen source, pure and rapeseed-biodiesel crude glycerol were analyzed comparatively as carbon sources for fermentation at 20 °C. Flask cultivations yielded 2.2 g/L and 2 g/L PHA at 120 h, respectively, with molecular weights of 428,629 g/mol and 81,515 g/mol. Bioreactor batch cultivation doubled biomass accumulation (10 g/L and 13.2 g/L) in 48 h, with a PHA productivity of 0.133 g/(L·h) and 0.05 g/(L·h). Thus, phenotypic and genomic analyses determined the successful use of *Photobacterium ganghwense* C2.2 for PHA production using urea and crude glycerol and 20 g/L NaCl, without pH adjustment, providing the basis for a viable fermentation process.

## 1. Introduction

To overcome the detrimental effects of the “Plastic Age” in which we live, it is imperative to finally switch from petrol-based plastics to bio-based alternatives. In Europe, almost 26 million metric tons of plastic waste are generated annually, and 80–85% of marine litter on European beaches is plastic [1]. Polyhydroxyalkanoates (PHA) are a bio-based, biodegradable, and biocompatible plastic alternative sourced from various microorganisms, which synthesize and accumulate it under limited nutritional conditions [2]. The most studied form of short-chain-length PHA (sclPHA) is poly(3-hydroxybutyrate) (P(3HB)), which has physicochemical properties similar to a widely used plastic, polypropylene [3].

PHA-producing microorganisms are key elements in moving towards a circular economy. In recent years, marine bacteria have been of great interest in the PHA research community [4] due to their ability to grow optimally at higher NaCl concentrations and lower temperatures. These properties can diminish the cost of fermentation strategies, as high salinity reduces the chance of contaminated cultures, and lower temperatures reduce the heating cost of production.

Crude glycerol is an ideal carbon source for PHA production, as it is an industrial by-product of the biodiesel industry, produced in large quantities (100 g per 1 kg of biodiesel) [5,6], which cannot be used in other processes without additional purification. As it has a high carbon–low nitrogen content—the most common condition for microbial PHA accumulation—unpurified, crude glycerolis an ideal base for PHA production. The use of crude glycerol as a sole carbon source for the production of PHA has been previously studied and reviewed [5].

*Photobacterium ganghwense* strain C2.2 (DSM 109767) is a marine bacterium isolated from Black Sea shore sediment, which has been shown to produce PHA using pure glycerol as a sole carbon source [7]. Although the genus has not been heavily studied from a PHA production perspective, we have previously obtained a complete genome for the strain. In this paper, we combined in-depth genomic analyses with phenotypic tests and shake flask cultivations to establish the choice of carbon and nitrogen sources for PHA production. The information gathered provided insights for the use of the strain for PHA accumulation through fermenter cultivations. Locally sourced crude glycerol was chemically characterized and employed as a sole carbon source for PHA production. The resulting biomass was analyzed through gas chromatography and size exclusion chromatography (SEC) in order to characterize the produced polymer. The strain was proven to have great metabolic versatility in PHA production conditions and was able to accumulate high amounts of PHA and biomass using both pure and crude glycerol as carbon sources, along with urea as a low-cost nitrogen source.

## 2. Results and Discussions

### 2.1. Cultivation Conditions for PHA Accumulation for Strain C2.2

#### 2.1.1. Establishing a Suitable Nitrogen Source for PHA Production

We tested several frequently used carbon sources for PHA production, using synthetic seawater minimal medium (ASW) containing NH_4_Cl (0.2% *w*/*v*) as a sole nitrogen source. The strain performed similarly when using laboratory-grade simple sugars and glycerol, reaching an optical density at 600 nm wavelength (OD_600_) of ~1.45 (Appendix A). Glycerol was chosen for a Biolog test using plate PM3, which contains a wide array of nitrogen sources. Based on OD_600_ values, the strain exhibited the most growth when supplied with nucleotides and nucleosides such as adenine and adenosine (Appendix A), as well as inosine, uric acid, and xanthosine, reaching values up to 2.51. Amino acids and dipeptides also increased cell growth, out of which the L-arginine well reached a peak optical density at 590 nm (OD_590_) of 2.54. Out of the simple nitrogen sources present in the plate, urea determined the best response (OD_590_ = 1.90) (Figure 1). For confirmation of this result, the cultivation was upscaled to shake flasks, using fructose, with various low-cost and readily available nitrogen sources: ammonium chloride, ammonium sulphate, ammonium nitrate, and urea.

The ammonia salts had no effect on the growth of the strain, in contrast with the existing literature, where differences were observed when the same ammonia salts were tested as nitrogen sources for PHA production using a strain of *Bacillus thuringiensis*, with ammonium chloride determining the best growth and polymer accumulation [8]. On the other hand, the addition of urea resulted in an almost sevenfold increase in optical density, from a peak mean OD_600_ = 1.49 for the ammonium chloride culture to OD_600_ = 10.30 for the urea culture (Appendix A). This prompted us to re-test the initial carbon sources using urea (0.2% *w*/*v*), which determined as much as a 27-fold increase in OD_600_ values, which was observed for the glycerol culture (OD_600_ = 38.7 at 96 h) (Appendix A). Cell biomass was analyzed at the 120 h timepoint, the glycerol culture had reached a cell dry weight (CDW) of 3.48 g/L, and gas chromatography pulsed ionization discharge detection (GC-PDD) analyses indicated a PHA content of 50.5%, comparable with the simple carbohydrate-based cultivations (Appendix A). From a genomic standpoint, the ability to successfully assimilate and metabolize urea is supported by the annotation of a cluster comprised of 13 genes (Appendix A). This cluster is situated immediately upstream of the PHA synthesis gene cluster *phaBAPC*. The presence of the *ureA*, *ureB*, *ureC*, and *ureD* genes indicates that the synthesis of functional urease is possible. The *urtABCDE* genes are essential for urea uptake and transport, which are upregulated in nitrogen starvation conditions—specifically, the absence of NH_4_^+^ [9]. Urea has been tested as a cost-effective nitrogen source in other biotechnologically relevant studies—in the case of PHA production using a *Bacillus* sp. strain [10], as well as for lipid production using yeast *Yarrowia lipolytica* [11].

The increase in PHA production when using urea as a sole nitrogen source, coupled with the close proximity of the urea metabolization pathway genes upstream of the PHA synthesis gene cluster, indicates there could be an increase in the transcription rate of the *phaBAPC* gene cluster when urea is supplemented in the growth media, owing to the increased transcription of the urea transporter and urease-encoding genes. This hypothesis will be the subject of future research.

#### 2.1.2. Establishing a Suitable Carbon Source for PHA Production

In order to efficiently screen a larger variety of carbon sources, a Biolog test was performed using PM1 plates. *P. ganghwense* C2.2 was able to grow to the highest optical density values using most monosaccharides (Figure 1), as well as maltotriose (Appendix A). Glycerol and myo-inositol determined the highest values after 48 h of cultivation. Moderate growth was observed on various organic acids (Appendix A), with propionic acid standing out, as it is used as a supplement for the production of hydroxyvalerate-containing polyhydroxyalkanoates [12]. The strain was also able to grow using Tween 20, Tween 40, and Tween 80, indicating the ability to metabolize lauric acid (C12:0), palmitic acid (C16:0), and oleic acid (C18:1), respectively [13]. The ability to grow on most of the tested carbon and nitrogen sources is an indicator of the strain’s extraordinary metabolic activity, characteristic of chemoorganotrophic free-living bacteria [14]. 

Glycerol was chosen for further experiments, as we considered that sugars such as fructose, trehalose, N-acetyl-D-glucosamine, and inositol are high-value substrates, usable in many other production processes. Furthermore, glycerol is also a major byproduct of the biodiesel production process and, for use in other industrial processes, requires additional purification.

#### 2.1.3. Establishing Strain Tolerance to Abiotic Stress Conditions in PHA Production Conditions

Biolog assays were also carried out on PM9 plates in order to identify the range of osmolyte concentrations at which the strain can grow (Figure 1). *P. ganghwense* C2.2 managed to reach high optical density values on NaCl concentrations up to 9%, with its growth rate gradually decreasing from 6.5% to 9%. In the case of the urea-containing wells, growth was possible on much larger concentrations than those used, with an inhibiting effect observed at concentrations greater than 2%. The strain was able to grow well on most osmolytes (Appendix A), including NaCl 6% supplemented with various osmoprotectant molecules, high concentrations of ethylene glycol (up to 20%), and all available concentrations of sodium formate. Sodium lactate 1%, sodium formate 2%, as well as sodium phosphate 200 mM determined some of the highest optical density values after 24 h, which were maintained throughout the remaining cultivation period.

In order to evaluate strain pH tolerance, PM10 plates were used (Figure 1 and Appendix A). The strain was able to grow well on pH values ranging from 5.5 to 10, with pH 6, 7, 8, and 8.5 obtaining the highest optical density values, and values 9 and higher had a slight inhibiting effect. This wide range of tolerated pH values could prove to be advantageous, from a biotechnological perspective, by not requiring constant pH adjustment throughout the fermentation process.

For the most part, the description of the *P. ganghwense* species using type strain FR1311 [15] matches the description of the strain used in this study, with some differences: strain C2.2 has a wider range of optimal pH values for cell growth (pH 6–8.5), while the type strain exhibited optimal growth at pH 8–9 but was able to also grow in the pH 5–11 range; strain C2.2 tolerates concentrations of up to 9% NaCl, while strain FR1311 can only grow in the presence of, at most, 7% sodium chloride.

The *Photobacterium* genus, part of the *Vibrionaceae* family, has members known for their ability to tolerate various forms of abiotic stress: low temperature, high salinity, alkalinity, as well as high pressure. Most mechanisms employed by the members of this genus cover all four stress conditions and consist of intracellular low molecular mass osmolyte accumulation mechanisms (e.g., ectoine, glycine betaine, trehalose) as well as transporter-based mechanisms. *Vibrionaceae* tolerating high salinity have also been shown to withstand high pressure, low temperatures, and alkaline conditions [16]. Strain C2.2 meets the genomic requirements for glycine betaine synthesis through the biosynthesis I pathway (specific for Gram-negative bacteria) (Appendix A). Ectoine synthesis genes have also been previously annotated. The most common mechanism for the adaptation to alkaline pH values consists of modifying the pH locally, either through the production of acidic molecules or the excretion of H^+^. These mechanisms involve the action of ATP-binding cassette (ABC) transporters, as well as Na^+^/K^+^–H^+^ antiporters, encoded by *nha* genes, studied in *E. coli* and *Vibrio* sp. [17], which were also annotated in the strain genome (Appendix A). 

### 2.2. Composition of Crude Glycerol

Unprocessed crude glycerol was obtained from a Romanian biodiesel production plant, Mac Bio Diesel (S.C. Mac Farmacons S.R.L), which uses rapeseed oil as the main raw material for the production of fatty acid methyl esters (FAME). The safety data sheet described the crude glycerol as having a neutral pH (6–7.5), containing traces of potassium hydroxide and sodium hydroxide, traces of methanol (0.5%), water, and soaps. Analyses showed that the crude glycerol is composed of 53.8% glycerol (±0.52%), 23.23% (±0.31%) C_18_ fatty acids (FA), and 2.1% (±0.1%) C_16_ FA.

### 2.3. Comparison of the PHA Accumulation of Strain C2.2 Using Pure Glycerol and Crude Glycerol

To observe the effect that using crude glycerol as a sole carbon source would have on CDW, PHA accumulation, and polymer properties, we carried out shake-flask cultivations of *P. ganghwense* C2.2 using both pure (PG) and crude glycerol (CG). The strain produced larger amounts of biomass when grown on CG up to the fifth day of cultivation, after which the PG cultures continued to grow. A peak CDW value was reached at 96 h for the CG cultures, at 4.31 g/L (±0.118) (Figure 2). The two substrates determined similar PHA accumulations for the first five days of cultivation: the maximum PHA content for the crude glycerol cultivation was measured at 120 h (2.066 g/L (±0.273) (50.19% of the CDW)), while the pure glycerol culture yielded 2.191 g/L (±0.087) (63% of the CDW) after the same duration (Figure 2). Maximum PHA quantity for the pure glycerol culture was obtained after 7 days, at 5.2 g/L (±0.374). In existing studies, similar results were obtained in similar conditions (PHA type produced, CG composition, cultivation volume) using *Halomonas* sp. KM-1 [18], which produced 1.6 g/L PHB and 4.1 g/L CDW after 48 h. An important difference is that the cultivation temperature for the strain was 30 °C rather than 20 °C, the temperature used in this study. 

In the case of the CG cultivation, because the glycerol content of the CG was only 53.8%, the strain was given half of the total glycerol that the PG culture had, which justifies the reduced amount of PHA in this case. Upon the addition of the crude glycerol to the growth media, a light-colored floating precipitate formed, which another study attributed to the soaps present in the CG precipitating [19]. Other authors investigated this phenomenon and determined that the fatty acid fraction of crude glycerol enters a stable colloidal state between pH 11.1 and 6.4 due to the decrease below the pKa values for C_16_ and C_18_ fatty acids. pH values below 6.4 determined the aggregation of the colloid into a floating layer [20]. Both studies in which this phenomenon was observed chose to further purify the CG, this fraction having had an inhibiting effect on the bacterial culture [19]. We decided not to include any CG purification steps so as to not increase the cost of the carbon source used. 

#### 2.3.1. Fatty Acid Degradation

As far as the utilization of fatty acids as a carbon source is concerned, genes encoding for the generic fatty acid β-oxidation I pathway were annotated (Appendix A). The existence of five copies of the *fadD* gene, as well as other genes involved in the FAD pathway (four copies of *fabG*), is a trait common to gram-negative bacteria, which are known to have acquired FAD gene homologs, most likely through horizontal gene transfer, with the resulting enzymes possibly having different specificities [21]. 

#### 2.3.2. Methanol Tolerance

Seeing as the crude glycerol used in this experiment also has methanol in its composition, we searched for genes involved in metabolizing this alcohol, which is toxic for some bacterial species commonly employed in sclPHA production, such as *Cupriavidus necator* [22]. We identified a gene encoding for a catalase-peroxidase (Appendix A), a category of enzymes which often have a broad range of possible substrates and could catalyze the oxidation of methanol to formaldehyde [23]. The strain possesses genes that encode for enzymes in the formaldehyde oxidation II and VII pathways (Appendix A), which have formate as the final product and which can then be oxidized to CO_2_. 

#### 2.3.3. PHA Molecular Weight and Biomass Fatty Acid Content

Polymer molecular weight was analyzed over time through size exclusion chromatography (SEC) for both the PG and the CG cultivations (Figure 3). The highest PHB molecular weights (M_w_) were observed after the first 24 h of growth: 428,629 g/mol (±755) and 81,515 g/mol (±5866) for PG and CG, respectively, when the PHA formation process was at its beginning. These values decreased over time, and at the timepoint at which the highest PHA content was obtained (120 h), the PHA had an M_w_ of 318,038 g/mol (±14,148) and 21,457 g/mol (±2080) for PG and CG, respectively. Molecular mass homogeneity was also evaluated through the polydispersity index (PDI, the ratio of the average number molecular weight (M_n_) to M_w_). The polymer obtained through CG cultivation had a much lower PDI throughout the seven days (PDI at 72 h was the lowest obtained, at 1.53), indicating a decreased diversity in polymer length. The pure glycerol PHB maintained a relatively constant PDI—on average, a value of 2.83 (±0.52) throughout the seven days—with the polymer exhibiting a high degree of homogeneity after the first 24 h of cultivation, at 1.67, when the average molecular weight was also highest. Furthermore, the same behavior was observed for all biological replicates in each cultivation condition, indicating a consistent and reproducible mechanism for PHA molecular weight modulation.

Many factors are involved in the molecular weight of the PHA a strain produces, some of which are strain-specific, PHA synthase-specific (optimal enzymatic activity pH, PHA synthase concentration), as well as the simultaneous action of PHA depolymerases on the produced polymer [24], although previous work shows that no intracellular PHA depolymerase gene was annotated in the C2.2 strain genome [7]. The factor we consider to be the most important in the case of the lower molecular weight of the PHA obtained from the tested carbon sources is the presence of chain termination (CT) agents, which act as terminator molecules for the polymer chains, their hydroxyl groups binding covalently to the carboxy-terminus of the, in this case, P(3HB) chain. Examples of such molecules are glycerol, whose effects have been studied previously, an increase in glycerol content determining the reduction in polymer molecular weight [25,26]. The molecular weight of the PG-based PHB is within the range described in other studies, summarized by [27]. Other known CT agents are alcohols such as ethanol, propanol, butanol, and a commonly encountered component of crude glycerol, methanol [28]. Ashby and co-authors, using a PHB-producing *Pseudomonas* strain, simulated the effect of the presence of methanol, representing, proportionally, up to 50% of a crude glycerol, and observed the presence of methoxy groups at the end of PHA chains. The addition of methanol resulted in a reduction in M_w_ of up to 82%, and methoxy groups increased in number throughout the cultivation period.

The most easily observable differences between the two carbon sources used is that, for the crude glycerol cultivation, 50% less glycerol is supplied, and the crude glycerol also supplies the strain with approximately 5 g/L FAs, as well as traces of methanol. The addition of these fatty acids and methanol to the growth media could justify the drastic reduction in polymer length. When looking at cell growth, at the 96 h mark, the CDW of the CG culture is almost 1 g/L higher, indicating that the composition of crude glycerol does not impact it negatively. Similarly, PHA accumulation was reduced by only ~13% when using CG compared to PG.

Because the average molecular weight of PHB dictates its thermo-mechanical properties (such as elasticity, tensile strength and crystallinity), high molecular weight (HMW) and Ultra-HMW PHB are the most desirable forms. There are multiple uses for low-molecular-weight (LMW) PHB (<1000 kDa) as well. LMW-PHB can also be an ideal “platform chemical”, as it is easier to depolymerize through chemical, enzymatic, and thermal treatments, without the need for the cell release of the polymer. Thus far, it has been used as a raw material for the production of compounds such as propene, crotonic acid, isocrotonic acid, methyl crotonate, methyl acrylate, cyclic/linear oligomers, and hydrocarbon oil, and its monomers are valuable molecules in and of themselves [29]. Similarly, 3-hydroxyalkanoic methyl esters are suitable as additives in biodiesel [3], and PHB can also be converted to a nearly zero-waste biofuel, with 99% carbon utilization and a 91.5% energy recovery rate [30]. PHB homopolymers have also proven to be effective as biocontrol agents [31], as well as in drug delivery and tissue engineering [31,32], while 3HB monomers and 3-hydroxybutyric methyl esters exhibit anti-osteoporosis and memory enhancement effects, respectively [32].

Along with the previous characterizations, we proceeded to quantify the major fatty acids present in the freeze-dried biomass for both types of cultivations (Figure 4). Unsaturated C_13_, C_16_, and C_18_ fatty acids, which varied in quantity throughout the cultivation period, were identified. These concentrations remained low throughout the entire cultivation period for the PG culture, while for the CG culture, a shift in concentrations was observed. The most significant difference over time was observed for C_18_, which had two accumulation cycles: it was initially up to 48 h (3.877% ± 1.074), then decreased by 72 h, and then proceeded to increase again up to a peak value of 9.416% (±1.235) after seven days had elapsed. C_16_ and C_13_ concentrations also increased over time, reaching peak values of 1.464% and 0.49%, respectively. The beginning of the second C_18_ accumulation cycle also coincides with the decrease in PHA molecular weight from the 72 h mark, indicating a possible interaction between PHA elongation and fatty acid accumulation, either at the metabolic (e.g., substrate competition) or the chemical interaction (e.g., end-capping) level. Observing that such a large portion of the CDW is represented by fatty acids (at most, 11.37% after 168 h, and 3.79% after 96 h) indicates that the PHA production process can also be of use in increasing the recovery of fatty acids from the crude glycerol. Furthermore, as both PHA and fatty acids are soluble in non-polar solvents, this could be a boon for the use of this strain as a producer of raw materials for biofuel production [33].

#### 2.3.4. Fatty Acid Biosynthesis 

The complete type II Fatty acid synthesis (FAS) pathway was annotated for the strain (Appendix A). Four *fadD* long-chain acyl-CoA synthetase genes, also involved in long/very long chain acyl-CoA synthesis, were identified. A gene encoding for a desaturase, stearoyl-CoA desaturase, was annotated, whose enzyme catalyzes the formation of oleoyl-CoA, palmitoleoyl-CoA, and monounsaturated C_18_ and C_16_ fatty acids. Possible products from these pathways include palmitoleic acid, oleic acid, lauric acid, stearic acid, and gondoic acid [34]. No polyunsaturated fatty acid (PUFA) synthesis genes were identified in the genome.

The *Photobacterium* genus is characterized as having C16:1 and C16:0 as the dominant fatty acids [35], but for the *P. ganghwense* species, C18:1 was determined as a major fraction of membrane fatty acids (29.6%), followed by C16:1 (27.9%) and C16:0 (21%) [15]. The cellular fatty acid content is subject to multiple forms of modulation (e.g., modifying chain length, degree of unsaturation, wax ester formation) as a response to multiple factors, including low temperatures, the substrate, the pH, and high pressure [36,37,38]. Significant correlations have been found between the modulation of the C16:0, C16:1, and C18:1 linear fatty acids present in bacterial cell membranes and the cultivation temperature: the C18:1 and C16:0 contents are greater at higher temperatures, and the C16:0 content increases at lower temperatures [36]. In the case of some *Vibrio parahaemolyticus* strains, in cold stress conditions, the 16:0 content decreased, while 16:1 and 18:1 were significantly increased [39]. This variation occurs because the cell membrane requires a specific degree of flexibility, which is adjusted by incorporating fatty acids that have varying melting points, a phenomenon termed homeoviscuous adaptation.

### 2.4. Comparison of PHB Accumulation in Batch Fermentation between Pure and Crude Glycerol

For the batch fermentations, the same conditions as those for the flask cultivation experiments were used, the only differences being a higher agitation speed and a doubling of the initial cell quantity. The CDW generated by *P. ganghwense C2.2* (Figure 5) reached 13.55 g/L (±2.15) after only 48 h of using CG as the carbon source and was maintained until 96 h. The biomass obtained at the 48 h point was threefold greater than the maximum CDW obtained for the same carbon source at smaller volumes. In the case of the PG batch experiment, the maximum CDW was achieved after 60 h of cultivation, with a biomass yield of 10.38 g/L (±0.47), representing more than double the amount of the flask cultivation in less than half the time. The ratio of carbon source to biomass yield was 0.66 and 0.52 for CG and PG, respectively, resulting in a high biomass incorporation ratio compared to another marine bacterium, *Salinivibrio* sp. M318, which had 0.5 in similar conditions [40]. The accumulation of PHB begins in the initial exponential phase and continues to increase until 48 h to 60 h, but the maximum amount of polymer was obtained at 36 h, with a value of 1.813 g/L of PHB and a mass fraction of 0.121, which led to a productivity of 0.05 g/(L·h) using crude glycerol. The biomass accumulation from pure glycerol, on the other hand, reached 9.76 g/L (±0.86) CDW at 48 h and contained a higher amount of PHB, 6.4 g/L (±0.55), 65 wt%, leading to a productivity of 0.133 g/L/h in unoptimized conditions. At the 48 h timepoint, the batch cultivations yielded a ratio of carbon source to PHB yield of 0.069 g PHB/g CG and 0.32 g PHB/g PG for the two types of batch cultivations. The ratio obtained for the PG fermentation was similar to what was recently obtained using *Photobacterium* sp. TLY01—0.31 g PHB/g glycerol—after 136 h of fed-batch cultivation at 30 °C [41]. These results are promising and indicate the need for further optimizations, since the PHB content in the cells increased from 43 wt% in the flask cultivation after 144 h of incubation to 65 wt% after only 48 h of bioreactor operation. Regarding pH, although both types of cultivation had an initial pH = 8.5, owing to the use of urea as a nitrogen source, pH values varied throughout the first 48 h of the fermentation process, reaching minimum values of 5.6 and 7.3, and pH 8 and 8.6 were recorded at 48 h for PG and CG, respectively. Thus, the overall setting of the cultivation was favorable for cell growth and biomass accumulation. Even though, in the case of CG, the batch cultivation did not provide such a large increase in volumetric productivity, the strain’s ability to produce such large amounts of biomass represents an excellent starting point for further optimizations, through which we believe that the strain could reach PHA content values at least matching those obtained in flask cultivations, but in less time.

The present results are in line with previous reports of glycerol as a preferred carbon source for PHB biosynthesis for marine bacteria. Such an example is *Vibrio harveyi* MCCB 284 [42], the bacterial strain studied for PHA production, which is most closely related to *Photobacterium ganghwense*, having had a productivity of 0.017 g/(L·h) on pure glycerol after optimization procedures. Similar results were obtained with *Burkholderia glumae* MA13, using crude glycerol from a biodiesel plant, which produced 1.96 g/L PHA from the 3.4 g/L of CDW after 72 h in the small volume experiments, a significant difference from our experiments being the higher pure glycerol content of the substrate (72%) [43]. Up-scaled experiments (i.e., fed-batch cultivation) using the same strain, which reached more than 9 g/L PHA and 14 g/L biomass, gives us a strong motivation for further optimizations of the large-volume cultivation process using *P. ganghwense* C2.2. According to Koller and Obruca (2022) [5], it is difficult to directly compare cultivation strategies for the production of PHA from crude glycerol, owing to the diversity of the bacterial strains employed, the different compositions and purities of the crude/industrial glycerol used, as well as a multitude of other cultivation parameters which vary at flask or up-scaled levels. So, in the end, productivity and overall costs will be the most important factors determining industrial viability.

## 3. Materials and Methods

### 3.1. Biolog Phenotypic Microarray Tests

Phenotype Microarrays (Biolog, Hayward, CA, USA) were used for the characterization of *P. ganghwense* C2.2 (DSM 109767) with regard to the usage pf carbon (PM1 plates) and nitrogen (PM3 plates) sources, osmolyte requirements (PM9 plates), and pH tolerance (PM10 plates). Synthetic seawater minimal medium (ASW) [44] without NH_4_Cl was the basal medium for the microarrays, supplemented with: 0.2% (*w*/*v*) urea for PM1; 2% (*w*/*v*) glycerol for PM3; 0.2% (*w*/*v*) urea, 2% (*w*/*v*) glycerol, and no NaCl for PM9; and 0.2% (*w*/*v*) urea, 2% (*w*/*v*) glycerol, 2% (*w*/*v*) NaCl, and pH adjusted to 7 before inoculation for PM10. For the PM array inoculation, the strain was grown on ASW supplemented with 1 g/L yeast extract and 5 g/L peptone (designated as “marine peptone”—MP) agar [7] for 24 h at 20 °C. Transmittance values were adjusted to 65%, and Biolog Redox Dye Mix A (#74221) was added, with a dilution factor of 100. A working volume of 100 µL was used in each well. Plates were incubated at 20 °C, and the optical density at 590 nm (OD_590_) was measured at 24 h intervals using the Biolog Microstation Reader (Biolog, Hayward, CA, USA). For data analysis, the OD_590_ values measured at the time of inoculation were subtracted from each of the following readings. Heat maps were generated using a custom python script.

### 3.2. Metabolic Pathway Identification

The strain’s complete genome (NCBI RefSeq:GCF_017329545.1) was annotated using gene ontology databases and associated software: KEGG database (accessed on 30 May 2021) [45], GhostKoala v2.2 (Kanehisa Laboratories, Kyoto, Japan) [46], and the KEGG Mapper Reconstruct tool [47]; EggNOG database (accessed on 6 June 2021) [48], the online server for emapper v2.1.4 (EMBL Computational Biology Group, Heidelberg, Germany) [49]; MetaCyc database (acessed on 25 June 2021) [50]; and PathwayTools version 25.0 (SRI International, Menlo Park, CA, USA) [51]. BLASTp searches against the UniProt database (accessed on 11 November 2021) [52] were used for validating the annotations.

### 3.3. Shake Flask Cultivation

The inoculum for the shake flask cultivation experiments was prepared by a 24 h cultivation on MP at 20 °C and 150 rpm. The inoculum was added to the experimental flasks up to a starting optical density at 600 nm (OD_600_) of 0.2 in 100 mL ASW (without NH_4_Cl and yeast extract) in 250 mL flasks. Culture proper aeration was ensured by using ePTFE membrane caps. Cultivation media were supplemented with 2% (*w*/*v*) carbon source and 0.2% (*w*/*v*) nitrogen source and incubated at 20 °C and 150 rpm. OD_600_, pH, and qualitative PHA granule accumulation were evaluated on a daily basis. PHA granule accumulation was assessed by preparing wet-mount slides from 1 mL culture, stained with 40 µL Nile Red solution (in ethanol, 40 µg/mL) and viewed through epifluorescence microscopy (Zeiss Axioplan, Carl Zeiss, Jena, Germany; FS00 (λ_ex_: 545/±25 nm, λ_em_: 560–710 nm)). For cell dry weight (CDW) determination and PHA quantification, 40 mL of the cell culture was centrifuged at 11,500× *g* for 5 min, and the resulting pellets were freeze-dried (−55°C, CHRIST ALPHA1-2 LD PLUS, Fisher Scientific, Hampton, NH, USA) and weighed.

In order to test the various carbon and nitrogen sources suitable for PHA production at larger volumes, flask cultivations were carried out using the following substrates: glycerol, fructose, glucose, galactose, and molasses, using NH_4_Cl as a sole nitrogen source. Various nitrogen sources were then tested using fructose as a carbon source: NH_4_Cl, (NH_4_)_2_SO_4_, urea, and NH_4_NO_3_. Cultivations were then carried out using urea as a nitrogen source and the previously mentioned carbon sources. After five days, samples were taken, freeze-dried, and used for CDW and PHA content evaluation.

Shake flask experiments comparing the use of pure glycerol (PG) and crude glycerol (CG), using urea as a nitrogen source, were also of interest. PG cultivation freeze-dried samples were obtained in a previous study [7] and stored at −20 °C. Cultivation using crude glycerol was carried out for seven days, and, in order to circumvent the depletion of bivalent ions (Mg^2+^, Zn^2+^) through the formation of soaps [53], we doubled the amount of their respective salts (MgCl_2_:6H_2_O, CaCl_2_:2H_2_O), as well as the microelement solution. Sterile crude glycerol was prepared by a 1:1 dilution with deionized water, followed by autoclave sterilization.

### 3.4. Bioreactor Cultivation Experiments

Scaled-up PHA production from PG and CG was performed on a 1.5 L BIOSTAT A Plus (Sartorius, Göttingen, Germany) bioreactor, with a double jacket vessel connected to a water bath as a cooling module (LT ecocool 100, Grant Instruments, Shepreth, UK). Batch cultivation experiments were conducted at 20 °C with a starting pH of 8.5 and aeration at 0.06 vvm at 500 rpm (using a Rushton turbine). The inoculum was added to the batch cultivation vessel to an initial OD_600_ = 0.5. The production media and the added supplements used were the same as those in the flask cultivations. The culture pH was not adjusted throughout the cultivations. For OD_600_, CDW, and PHA quantification and fluorescence microscopy evaluation, 45 mL samples were taken every 12 h.

### 3.5. Quantification of PHA and Fatty Acids by Gas Chromatography-Flame Ionization Detector (GC-FID)

The amount of PHA and fatty acids in the freeze-dried bacterial biomass obtained from the pure and crude glycerol-supplemented shake-flask cultivations was quantified through a GC-FID, following a methanolysis treatment in pre-mixed 1:1 (vol/vol) chloroform and methanol containing 15% concentrated sulphuric acid. Approximately 10–20 mg of freeze-dried cells was added to 3 mL of the methanolysis mixture containing 0.2% benzoic acid as the internal standard and reacted for 2 h at 100 °C. Tridecanoic, palmitic, and stearic acids (≥99%, Sigma-Aldrich, St. Louis, MI, USA) and PHBV pellet (12 mol% HV, Sigma-Aldrich) were used as external standards. The reaction mixture was cooled down to room temperature, and then 1.5 mL of Milli-Q water was added to each vial in order to form phase separation. The organic phase was carefully transferred into GC analysis vials. The samples were analyzed by an Agilent 7820A GC-FID system (Agilent Technologies, Santa Clara, CA, USA) equipped with a 30 m long by 0.32 mm i.d. DB-5MS column with 0.25 μm film thickness. The injection volume was set to 1 μL, with a split ratio of 10:1, and the injector temperature was set to 300 °C. The temperature program was set to 50 °C, which was held for 4 min, followed by a linear increase to 200 °C at a rate of 15 °C/min and then an increase to 300 °C at a rate of 30 °C/min, which was held for 8 min. Hydrogen was used as carrier gas at a 1 mL/min flow, with a column head overpressure of 43 kPa. Two technical replicates for each PHA extraction (for each of the two biological replicates per timepoint) were injected.

### 3.6. Quantification of PHA by Gas Chromatography-Pulsed Discharge Ionization Detector (GC-PDD)

The total amount and composition of PHA in the biomass obtained from the fermentor cultivations and the initial screening shake-flask cultivations were determined following methanolysis of freeze-dried, ground samples. Approximately 20 mg of dry pellet was subjected to methanolysis with 1.5% sulfuric acid/methanol (2 mL) and chloroform (3 mL) at 100 °C for 72 h in screw-capped test tubes (#261351258, DWK Life Sciences, Mainz, Germany). Benzoic acid (2 mg/sample) was used as an internal standard. Three different amounts of poly ((R)-3-hydroxybutyric acid) (PHB) (#363502, Sigma-Aldrich) were used as standards to create a calibration curve. After methanolysis, 2 mL of ammonia solution (12.5%) was added to separate the organic and aqueous phases. The organic phase containing methyl ester derivatives was analyzed by a GC system (Varian 3800) equipped with an He Pulsed Discharge Detector (Vici) and a CPSIL 5CB column (Varian) (100% dimethylpolysiloxane, 30 m, 0.25 mm diameter, 2 μm film thickness). The operating conditions were: 1 μL sample was injected with 6.0 purity helium as a carrier gas, 13.81 psi, total flow of 9 mL/min, column flow of 1 mL/min, purge flow of 3.0 mL/min, temperature increment of 10 °C/min from 50 °C to 220 °C, injector temperature of 250 °C, and detector temperature of 250 °C. Data analysis was performed using Varian Star Workstation 6.9.

### 3.7. Crude Glycerol Composition Analysis by GC-FID

The amount of glycerol in the crude glycerol sample was analyzed by using the GC-FID method. Approximately 10–20 mg of the sample was added to GC vials and diluted by the addition of 1 mL chloroform. The samples were injected, without prior preparation, to an Agilent 6890N GC-FID system (Agilent Technologies) equipped with a 30 m long by 0.25 mm i.d. HP-INNOWAX column with a 0.25 μm film thickness. The injection volume was set to 1 μL with a splitless mode, and the injector temperature was set to 300 °C. The temperature program was set to 150 °C and held for 10 min, followed by a linear increase to 180 °C at a rate of 5 °C/min, which was held for 10 min, and then by an increase to 260 °C at a rate of 5 °C/min, with a 2 min hold. Hydrogen was used as the carrier gas at a 1.1 mL/min flow, with a column head overpressure of 122 kPa. Three technical replicates were injected. The fatty acids content was determined as in the methanolysis method above.

### 3.8. Polymer Extracts Preparation and Molecular Weight Determination by Size Exclusion Chromatography (SEC)

Polymer samples were prepared by adding approximately 50–100 mg of lyophilized cells into 5 mL pressure-resistant screw cap tubes (Duran) containing 3 mL of chloroform (VWR, USA). The samples were heated for 4 h at 80 °C, cooled down to room temperature, and filtered through 0.2 µm nylon syringe filters (Thermo Fisher) into 20 mL glass vials. The polymer was precipitated by the addition of 3 vol. equivalents of ice-cold methanol (VWR). Polymer precipitates were collected on Millipore 0.2 µm Nylon membrane filters (Merck, Darmstadt, Germany) with an additional 5 mL methanol washing step and dried overnight at room temperature in a vacuum oven (Gallenkamp, London, UK). The extracted polymer samples were then re-dissolved in chloroform at an approximate concertation of 5 mg/mL and transferred to LC analysis vials. The molecular weight of the polymer extracts was measured by a size exclusion chromatography system consisting of a G1311A HPLC pump (Agilent Technologies, USA) and two T6000M (Malvern Panalytical, Malvern, UK) columns in series, with chloroform as the eluent. The detection was carried out by using the Refracting Index (RI) and multiangle light scattering detectors (Malvern Panalytical). The temperature in the autosampler, the columns, and the detector was maintained at 30 °C, with a flowrate of 1 mL/min. The MALS data were evaluated using the Debye model with the first-degree polynomial in Omnisec 11.20 software (Malvern Panalytical). The detectors were calibrated with the polystyrene standards PSS-ps200k and PSS-psb310k (PSS, Pleasanton, CA, USA). Data analysis and visualization were aided by Daniel’s XL Toolbox add-in for Excel, version 7.3.4, by Daniel Kraus, Würzburg, Germany (www.xltoolbox.net (accessed on 20 June 2021)).

## 4. Conclusions

This work presents the marine bacterium *Photobacterium ganghwense* C2.2 as a promising PHA producer. A comprehensive picture of the strain’s metabolic capabilities in relation to PHA production was determined through correlating genomic and phenotypic data. The metabolic pathways identified indicated the suitability of urea as a nitrogen source, the strain’s ability to assimilate and tolerate crude glycerol contaminants, as well as a tolerance for pH and salinity variation in a fermentation process. Comparisons of pure and crude glycerol-based cultivations outline the influence that the industrial byproduct has on biomass quantity and composition, PHA content, and molecular weight. Flask and bioreactor batch cultivations demonstrated the strain ability for reaching a high PHA and biomass accumulation. Using unpurified crude glycerol and urea as a low-cost nitrogen source and maintaining a low cultivation temperature of 20 °C, without additional pH adjustment, constitutes the basis for a cost-effective biotechnological process.

## Figures and Tables

**Figure 1 ijms-23-13754-f001:**
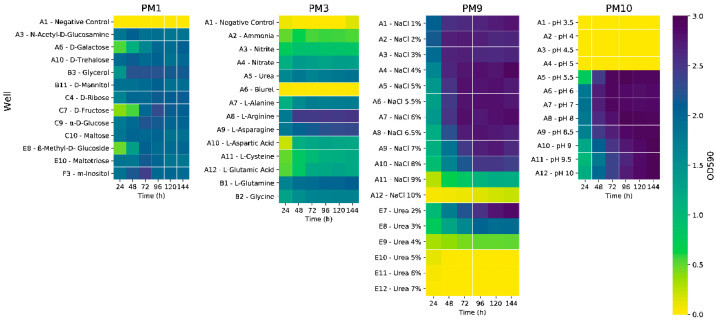
Heatmaps depicting the growth (OD_590_ values) of strain C2.2 in selected wells of Biolog plates PM1, PM3, PM9, and PM10. OD_590_ values determined at 24 h intervals.

**Figure 2 ijms-23-13754-f002:**
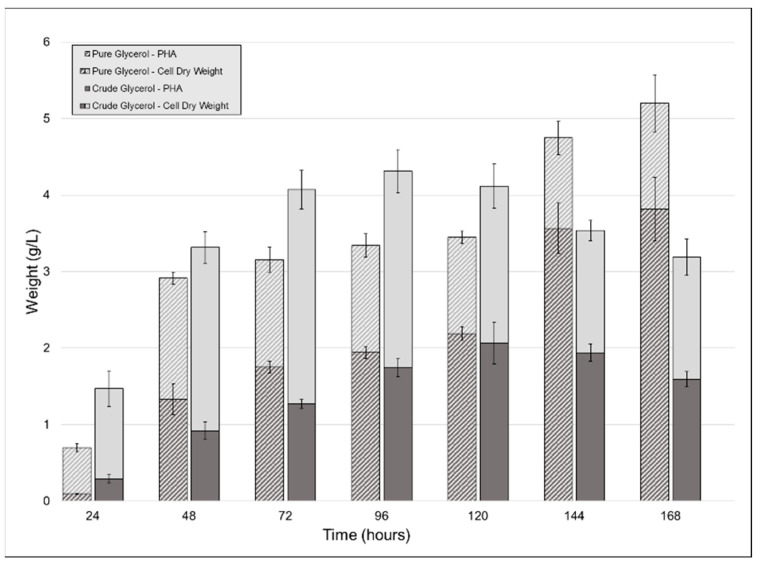
Biomass and polyhydroxyalkanoates (PHA) obtained from the cultivation of strain C2.2 using crude glycerol and pure glycerol as the sole carbon sources throughout a seven-day shake-flask cultivation. Error bars represent the standard deviation of biological and technical replicates.

**Figure 3 ijms-23-13754-f003:**
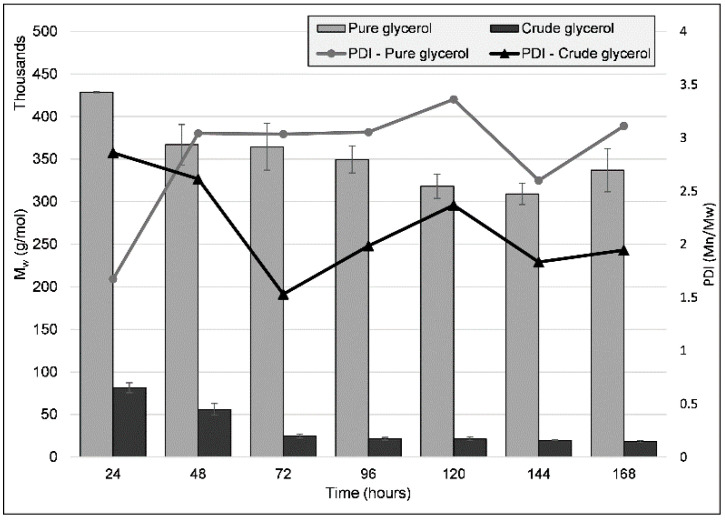
Molecular weight (M_w_) and PDI (polydispersity index, the ratio of the average number molecular weight (M_n_) to M_w_) values for the PHA obtained from the cultivation of strain C2.2 using pure glycerol and crude glycerol as sole carbon sources, respectively, throughout a seven-day shake-flask cultivation. Error bars represent the standard deviation of biological and technical replicates.

**Figure 4 ijms-23-13754-f004:**
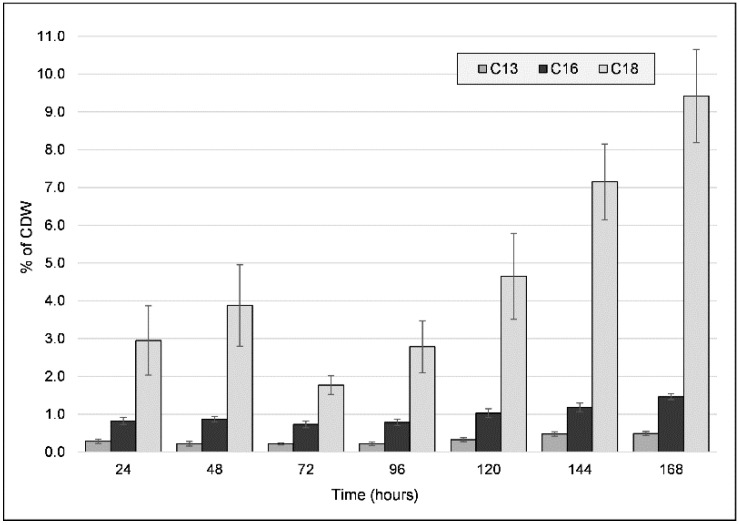
Fatty acid (C_13_, C_16_, C_18_ fatty acids) content (wt% of cell dry weight (CDW)) of freeze-dried biomass obtained from the cultivation of strain C2.2 using crude glycerol as a sole carbon source throughout a seven-day shake-flask cultivation. Error bars represent the standard deviation of biological and technical replicates.

**Figure 5 ijms-23-13754-f005:**
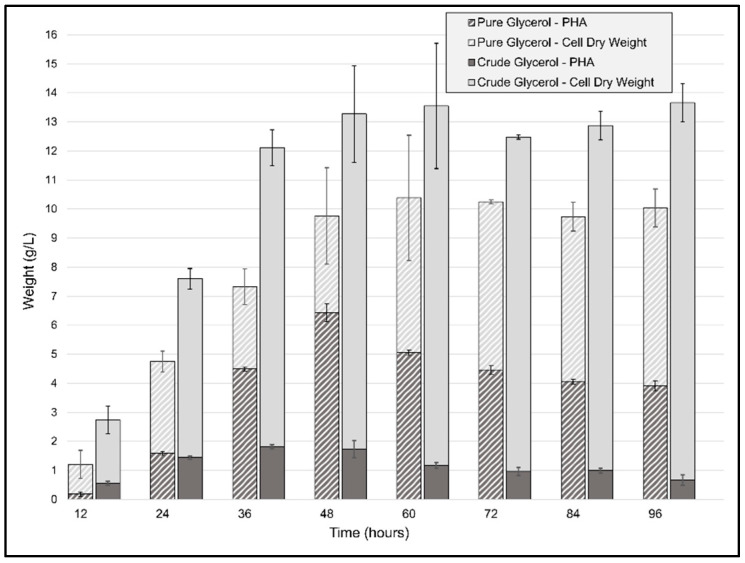
Cell dry weight (CDW) and polyhydroxyalkanoate biosynthesis profile of *P. ganghwense* in 2 L batch cultivations using 20 g/L pure glycerol and crude glycerol, respectively. Error bars represent the standard deviation of biological and technical replicates.

## Data Availability

The complete genome of *Photobacterium ganghwense* C2.2 (DSM 109767) is available on the NCBI RefSeq under the accession number GCF_017329545.1. The strain is available from the DSMZ–German Collection of Microorganisms and Cell Cultures under the accession number DSM 109767. The script for the Biolog data visualization is available on Github (https://github.com/danieltudosiu/optical_density_well_plate_heatmap accessed on 15 June 2022).

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
