# Peer review of "Revealing the Phenotypic and Genomic Background for PHA Production from Rapeseed-Biodiesel Crude Glycerol Using Photobacterium ganghwense C2.2"

_ijms, 2022, doi:10.3390/ijms232213754_

Round 1

Reviewer 1 Report

Journal.            IJMS (ISSN 1422-0067)       20221015

Manuscript ID.  ijms-1966453

Type.                Article

Title             Revealing phenotypic and genomic background for PHA pro-duction from rapeseed-biodiesel crude glycerol using Photo-bacterium ganghwense C2.2

Authors              Irina Lascu , Ana-Maria Tanase * , Piotr Jablonski , Iulia Chiciudean , Maria Irina Preda , Sorin Avramescu , Knut Irgum , Ileana Stoica 

Section.             Molecular Microbiology

Special Issue.    Molecular Advances in Microbial Metabolism

A few issues:

1.  Authors have concluded that 

Photobacterium ganghwense C2.2 for PHA production using urea and crude glycerol, 20 g/L NaCl, and without pH adjustment, providing the basis for a viable fermentation process.

The results further state that: 

The ammonia salts had no effect on the growth of the strain, but the addition of urea resulted in an almost sevenfold increase in optical density

Q:  A review of literature (Ref. https://doi.org/10.1016/j.ijbiomac.2015.03.046) shows that various Ammonium salts have different impact of PHA yield and characteristics.

It shows that addition of Ammonium Chloride leads  to maximum benefits.

     Thus, the authors need to justify their results in light of the previously published works.

     2. It appears that in this work only PHB production has been obtained.

     Q. Bioconversion of Glycerol to homopolymers (PHB) alone  is an inefficient and un-economical proposal.

          CoPolymers of PHA have many applications compared to homopolymers.

              Ref.: https://doi.org/10.1007/s12088-017-0651-7.

              Ref. : https://doi.org/10.1016/j.biortech.2021.124737

   It is suggested that the authors should re-analyze their samples for Copolymer production.

Author Response

The reviewers’ comments are in black, and our responses are in red. The line numbers (as visible when “All markup” is displayed) in the revised version of the manuscript are also in red.

The authors want to thank to the reviewer for their patience and time spent reading and evaluating the manuscript.

A few issues:

  1. Authors have concluded that 

“Photobacterium ganghwense C2.2 for PHA production using urea and crude glycerol, 20 g/L NaCl, and without pH adjustment, providing the basis for a viable fermentation process”.

The results further state that: 

The ammonia salts had no effect on the growth of the strain, but the addition of urea resulted in an almost sevenfold increase in optical density

Q:  A review of literature (Ref. https://doi.org/10.1016/j.ijbiomac.2015.03.046) shows that various Ammonium salts have different impact of PHA yield and characteristics.

It shows that addition of Ammonium Chloride leads to maximum benefits.

      Thus, the authors need to justify their results in light of the previously published works.

A: The statement that “The ammonia salts had no effect on the growth of the strain” was based on the measurement of optical density values, available in Figure S1.B, where all ammonia salts basically followed the same growth curve. We believe this behaviour to be species- and strain- specific, as a possible justification for the growth increase obtained from using urea was based on the close proximity of the urease and urea transport gene clusters to the phaCPAB cluster (mentioned from L102 to L113). We also added the following text as an additional discussion:

L90 –   “The ammonia salts had no effect on the growth of the strain, in contrast with existing literature, where differences were observed when the same ammonia salts were tested as nitrogen sources for PHA production using a strain of Bacillus thuringiensis, ammonium chloride determining the best growth and polymer accumulation [8].”

  1. It appears that in this work only PHB production has been obtained. Bioconversion of Glycerol to homopolymers (PHB) alone  is an inefficient and un-economical proposal.

          CoPolymers of PHA have many applications compared to homopolymers.

              Ref.: https://doi.org/10.1007/s12088-017-0651-7.

              Ref.: https://doi.org/10.1016/j.biortech.2021.124737

   It is suggested that the authors should re-analyze their samples for Copolymer production.

A: Unfortunately, it is not possible for us to re-analyze the samples for copolymer production, but we have tried accentuate that, although the PHB produced by the strain is not viable as a standalone plastic material, the homopolymers and its monomers are still valuable for numerous applications. We have already mentioned the utility of LMW-PHB as both a “platform chemical” and as a substrate for the production of biofuels (L278). Furthermore, we have added the following phrase in support of this idea:

L287 – “PHB homopolymers have also proven to be effective as biocontrol agents [32], as well as in drug delivery and tissue engineering [32,33], while 3HB monomers and 3-hydroxybutyric methyl esters exhibiting anti-osteoporosis and memory enhancement effects, respectively [33].”

We would also like to say that the focus of this paper was on the analysis of a PHA production process based on the use of crude glycerol, as cheap carbon source for economically feasible process and the effects of various parameters regarding biomass and PHA accumulation. The production of copolymers will be the subject of a future paper, through addition of volatile fatty acids such as propionate and valerate, see reference [42]   https://doi.org/10.1016/j.jece.2022.108342

Reviewer 2 Report

The manuscript of Lascu et al. presents the marine bacterium Photobacterium ganghwense C2.2 as a promising PHA producer by linking a wide range of characterization methods: metabolic pathway annotation from the strain’s complete genome, high-throughput phenotypic tests, and biomass analyses, through plate-based assays, flask and bioreactor cultivations. Authors have been able to set up an inclusive relation of C2.2 to PHA production through correlating genomic and phenotypic data. Comparisons of pure and crude glycerol-based cultivations and its effect on biomass measure and composition, PHA content and molecular weight is very well studied.

Author Response

We thank the reviewer for their time and appreciation.